Construction and validation of m6A-related diagnostic model for psoriasis

Liu Jing 1 2
Wang Youlin 1
Sheng Yu 2
Cai Limin 2
Wang Yongchen yongchenwang@163.com 1 3
1 Department of Dermatology, The Second Affiliated Hospital of Harbin Medical University , Harbin , Heilongjiang , China
2 Department of Dermatology, The First Affiliated Hospital of Harbin Medical University , Harbin , Heilongjiang , China
3 General Practice Department, The Second Affiliated Hospital of Harbin Medical University , Harbin , Heilongjiang , China
Orlov Yuriy
Electronic publication date: 2024 Feb 29
Publication date: 2024
Volume: 12
Electronic Location ID: e17027
Received 2023 Oct 18; Accepted 2024 Feb 7
Copyright: ©2024 Liu et al.
Copyright year: 2024
Copyright holder: Liu et al.
License: This is an open access article distributed under the terms of the Creative Commons Attribution License, which permits unrestricted use, distribution, reproduction and adaptation in any medium and for any purpose provided that it is properly attributed. For attribution, the original author(s), title, publication source (PeerJ) and either DOI or URL of the article must be cited.
License URL: https://creativecommons.org/licenses/by/4.0/

Keywords: Psoriasis, m6A, Diagnostic model, Bioinformatics analysis

Funding: The research and innovation fund of the First Affiliated Hospital of Harbin Medical University No. 2023M10 This work was supported by the research and innovation fund of the First Affiliated Hospital of Harbin Medical University (No. 2023M10). The funders had no role in study design, data collection and analysis, decision to publish, or preparation of the manuscript.

==============================
Background

Psoriasis is a chronic immune-mediated inflammatory disease. N6-methyladenosine (m6A) is involved in numerous biological processes in both normal and diseased states. Herein, we aimed to explore the potential role of m6A regulators in the diagnosis of psoriasis and predict molecular mechanisms by which m6A regulators impact psoriasis.

Methods

GSE30999 (170 human skin tissue samples) and GSE13355 (180 human skin tissue samples) were downloaded as the training analysis dataset and validation dataset respectively. M6A-related genes were obtained from the literature and their expression levels in GSE30999 samples were measured to identify M6A-related DEGs between psoriasis lesions (LS) and non-lesional lesions (NL). We identified m6A-related DEGs using differential expression analysis and assessed their interactions through correlation analysis and network construction. A logistic regression analysis followed by LASSO optimization was employed to select m6A-related DEGs for the construction of a diagnostic model. The performance of the model was validated using support vector machine (SVM) methodology with sigmoid kernel function and extensive cross-validation. Additionally, the correlation between m6A-related DEGs and immune cell infiltration was analyzed, as well as the association of these DEGs with psoriasis subtypes. Functional analysis of the m6A-related DEGs included the construction of regulatory networks involving miRNAs, transcription factors (TFs), and small-molecule drugs. The m6A modification patterns were also explored by examining the gene expression differences between psoriasis subtypes and their enriched biological pathways. Finally, the expression of significant m6A regulators involved in the diagnostic model was examined by RT-qPCR.

Results

In this study, ten optimal m6A-related DEGs were identified, including FTO, IGF2BP2, METTL3, YTHDC1, ZC3H13, HNRNPC, IGF2BP3, LRPPRC, YTHDC2, and HNRNPA2B1. A diagnostic model based on these m6A-related DEGs was constructed, demonstrating high diagnostic accuracy with an area under the curve (AUC) in GSE30999 and GSE13355 of 0.974 and 0.730, respectively. Meanwhile, the expression level of m6A regulators verified by RT-qPCR was consistent with the results in GSE30999. The infiltration of activated mast cells and NK cells was significantly associated with all ten m6A-related DEGs in psoriasis. Among them, YTHDC1, HNRNPC, and FTO were targeted by most miRNAs and were regulated by nine related TFs. Therefore, patients may benefit from dorsomorphin and cyclosporine therapy. Between the two subgroups, 1,592 DEGs were identified, including LRPPRC and METTL3. These DEGs were predicted to be involved in neutrophil activation, cytokine-cytokine receptor interactions, and chemokine signaling pathways.

Conclusions

A diagnostic model based on ten m6A-related DEGs in patients with psoriasis was constructed, which may provide early diagnostic biomarkers and therapeutic targets for psoriasis.

Introduction

Psoriasis is a complex, genetically determined, chronic immune-mediated skin disease that affects at least 100 million people worldwide (Michalek, Loring & John, 2017). Psoriasis was recognized as a “chronic, non-communicable, painful, disfiguring, and disabling disease for which there is no cure” by member states in the World Health Assembly resolution in 2014. Psoriasis often seriously affects the patient’s quality of life because of its high prevalence, frequent recurrence, and chronic disease course. Compared with the general population, patients with psoriasis are more likely to experience depression and suicidal behaviors (Singh et al., 2017). The etiology and pathogenesis of psoriasis are not completely clear. Genetic susceptibility, environmental variables, and dysregulated immune responses play important roles in the disease (Griffiths et al., 2021). Although various drugs and other therapies are available clinically, psoriasis cannot be completely cured. Recently, owing to the continuous progress in molecular biology research methodologies, new prognostic markers have been discovered. However, more characteristic factors of psoriasis need to be identified and confirmed, which will be helpful for early diagnosis and targeted treatment.

Recent data have shown that pathological crosstalk between immune cells and keratinocytes can drive the progression of psoriasis (Sato, Ogawa & Okuyama, 2020). With the development of next-generation sequencing and bioinformatics, various genes involved in immune-related signaling pathways have been identified (Ghoreschi et al., 2021; Nogueira, Puig & Torres, 2020). N6-methyladenosine (m6A) is an essential epigenetic modification occurring at the sixth N of adenine in mRNA. In both noncoding and coding RNAs, m6A is recognized as the most prevalent RNA modification in eukaryotic species, is accepted as a crucial post-transcriptional regulator, and is involved in the pathogenesis of various immune-related diseases, including psoriasis (Bi et al., 2019). For example, m6A transcriptome-wide profiling performed by Wang et al. revealed that, in psoriasis, m6A affected 19.3–48.4% of differentially expressed transcripts (Wang & Jin, 2020). And they found the decrease of METTL3-mediated m6A methylation promoted the development of psoriasis in imiquimod-induced psoriasis-like mouse model (Wang, Huang & Jin, 2022). Yuan et al. (2023) revealed that the m6A modulator METTL3 and ALKBH5 regulated inflammation in psoriasis by modifying IL-17a transcripts in CD4+T cells. Thus, further identification of m6A-related modulators and their functions in psoriasis is essential.

For a more comprehensive understanding of the role of m6A modulators in the diagnostic biomarkers and subtype analysis of psoriasis, GSE30999 was downloaded as a training analysis dataset and GSE13355 as a validation dataset. Differentially expressed m6A-related genes in psoriatic lesions (LS) were analyzed. A diagnostic model based on m6A-related genes was established. Real-time quantitative polymerase chain reaction (RT-qPCR) experiments further validated significant m6A genes, exhibiting consistent expression levels with the bioinformatics results. Furthermore, disease subtype, immune infiltration and other aspects were analyzed based on diagnostic m6A-related DEGs (Fig. 1). Finally, this study successfully constructed a diagnostic model that may provide potential diagnostic biomarkers and new therapeutic targets.

Figure 1 Flow chart of this study.

Materials & Methods

Data download

GSE30999 and GSE13355 were obtained from the GEO database (http://www.ncbi.nlm.nih.gov/geo/), and both were sequenced using the GPL570 [HG-U133_Plus_2] Affymetrix Human Genome U133 Plus 2.0 Array platform. A total of 170 human skin tissue samples were included in GSE30999, which comprised 85 paired lesional and non-lesional samples. GSE13355 included 180 human skin tissue samples obtained from 116 patients with psoriasis and 64 normal control samples. In the present study, GSE30999 was designed as a training analysis dataset and GSE13355 was designed as a validation dataset.

Screening of differentially expressed m6A-related genes

By reviewing the literature (Wang et al., 2021), we selected a total of 25 m6A-related genes. These m6A-related genes include nine readers (METTL3, METTL14, METTL16, WTAP, KIAA1499, RBM15, RBM15B, RBM1, ZC3H13), 14 writers (YTHDC1, YTHDC2, YTHDF1, YTHDF2, YTHDF3, IGF2BP1, IGF2BP2, IGF2BP3, HNRNPA2B1, HNRNPC, HNRNPG, RBMX, FMR1, LRPPRC) and two erasers (FTO, ALKBH5). The corresponding expression levels of m6A-related genes in samples from the GSE30999 dataset were determined. M6A-related differentially expressed genes (DEGs) between psoriasis lesions (LS) and non-lesional lesions (NL) were selected using the limma package (Version 3.34.7, https://bioconductor.org/packages/release/bioc/html/limma.html) in the R 3.6.1. Associations between m6A-related DEGs were assessed using the cor function (https://cran.r-project.org/web/packages/COR/index.html) in r 3.6.1 (R Core Team, 2019) and displayed on a heatmap using pheatmap. Gene Ontology (GO) function and Kyoto Encyclopedia of Genes and Genomes (KEGG) signal pathway enrichment analysis of m6A-related differentially expressed genes were further performed using ClusterProfiler (https://www.bioconductor.org/packages/release/bioc/html/clusterProfiler.html) in R 3.6.1. Finally, a protein-protein interaction network was constructed using STRING (Version 11, http://string-db.org/) and visualized using Cytoscape (Version 3.9.0, http://www.cytoscape.org/) (Szklarczyk et al., 2017). In these analyses, FDR<0.05 was set as the threshold.

Construction of a diagnostic model based on m6A-associated genes

Referring to the levels of m6A-related DEGs in GSE30999, single-factor logistic regression analysis was performed using rms in the R 3.6.1 (version 6.3-0, https://cran.r-project.org/web/packages/rms/index.html). M6A-related DEGs with p value less than 0.05 were included for further analysis. Based on the above analysis results, the final optimized m6A-related genes were explored using the LASSO algorithm with the lars package in the r 3.6.1 (Version 1.2, https://cran.r-project.org/web/packages/lars/index.html) The risk score for each sample was calculated based on the LASSO coefficient of the optimized m6A-related genes. The Kruskal-Wallis test was used to compare the risk scores between psoriasis and normal control samples.

Finally, the diagnostic model based on m6A-related DEGs was constructed. The disease diagnosis classifier based on m6A-related genes in the GSE30999 training set was constructed using the Support Vector Machine (SVM) method (Meyer et al., 2023) based on sigmoid kernel and 100-fold cross-validation. The Receiver operating characteristic (ROC) curve was then used to evaluate the effectiveness of the disease diagnosis model based on data from the GSE30999 training dataset and GSE13355 validation dataset using R 3.6.1, pROC package (Version 1.12.1, https://cran.r-project.org/web/packages/pROC/index.html).

Functional analysis of m6A-related DEGs in the diagnostic model

Three m6A-related DEGs regulatory networks were constructed, including miRNA, TFs and small-molecule drugs. miRNAs associated with m6A-related genes in the diagnostic model were screened using the miRWalk 3.0 database (http://mirwalk.umm.uni-heidelberg.de/), and connection pairs labeled “validated” were defined as effective connection pairs. Finally, the miRNA-m6A-related DEGs regulatory network was visualized using Cytoscape. Information associated with the connection between TFs and regulatory target genes was downloaded from the Translational Regulatory Relationships Untraveled by Sentence-based Text mining Database (TRRUST, https://www.grnpedia.org/trrust) Han et al. (2018). M6A-related genes involved in regulatory diagnosis were selected to build the TF-m6A-related DEGs regulatory network. Small-molecule drugs related to m6A-related DEGs in the diagnostic model were screened using DrugBank (https://go.drugbank.com/drugs), and the connections between m6A-related genes and small-molecule drugs were displayed using Cytoscape Version 3.9.0.

Analysis of the correlation between m6A-related DEGs and immunity

According to the CIBERSORT algorithm (https://cibersort.stanford.edu/index.php) (Chen et al., 2018), the infiltration of immune cell types in samples involved in the GSE30999 dataset was evaluated. The Kruskal-Wallis test was used to compare the distribution of different immune cells in psoriasis and control groups. Finally, the correlation between the expression levels of m6A-related DEGs and immune cells was determined using the cor function. Interactions between m6A-related genes were determined using GeneMANIA (http://genemania.org/).

Analysis of sample subtypes based on a diagnostic m6A-related DEGs signature

Disease subtype analysis of the samples in GSE30999 was performed by ConsensusClusterPlus (http://www.bioconductor.org/packages/release/bioc/html/ConsensusClusterPlus.html) in the R 3.6.1 (Zhang et al., 2020). The m6A scores of samples in GSE30999 were evaluated using GSVA (Version 1.36.3, http://www.bioconductor.org/packages/release/bioc/html/GSVA.html) in the R 3.6.1 (Ye et al., 2019), and the differences in m6A scores between subtypes were compared using the Kruskal-Wallis test. Finally, the expression levels of m6A -related DEGs in different subtypes were displayed using a heatmap by pheatmap in the R 3.6.1 (Wang et al., 2014) (Version 1.0.8, https://cran.r-project.org/web/packages/pheatmap/index.html).

Correlation analysis between subtypes and immunity

The infiltration of immune cells in samples involved in GSE30999 was evaluated using CIBERSORT (Chen et al., 2018). The ESTIMATE, immune, and matrix scores in GSE30999 were calculated by the estimate package in the R 3.6.1 (Hu, Zhou & Zhu, 2019) (https://bioinformatics.mdanderson.org/estimate/rpackage.html). The Kruskal-Wallis test was used to compare the infiltration of immune cells and ESTIMATE scores in different subtypes. After, the expression levels of HLA family genes in GSE30999 were selected and the differences in the expression levels of HLA family genes among subtypes were compared using the inter-group t-test.

Recognition of m6A modification pattern

DEGs in subtypes were screened using the limma package, and FDR values less than 0.05 and —Log2FC—>0.5 were set as thresholds. GO biological processes and KEGG signaling pathways enriched by DEGs were selected using ClusterProfiler. Weighed gene coexpression network analysis (WGCNA) is a tool for identifying modules related to diseases and screening potential therapeutic targets or important pathogenic mechanisms for the disease. Modules significantly related to the psoriasis subtypes were screened by the WGCNA package in the R 3.6.1 (Langfelder & Horvath, 2008)(Version 1.61, https://cran.r-project.org/web/packages/WGCNA/index.html). The module division screening threshold was set as follows: cut height =0.995, and the module set contained at least 50 genes. The genes involved in the module with the highest correlation with the subtype were identified as candidate genes related to the subtype. Finally, a protein-protein interaction network of these genes was constructed using STRING and visualized using Cycloscape Version 3.9.0.

Human tissue sample collection

Skin tissues (including six psoriasis patients and six healthy controls) were recruited from the Department of Dermatology at the First Affiliated Hospital of Harbin Medical University. The clinical information of the patients is provided in Table S1. The psoriasis patients did not receive any treatment for at least one month before the skin biopsy. All participants provided written informed consent, and the study protocol was approved by the ethics committee of the First Affiliated Hospital of Harbin Medical University (No. 202325).

Animal model

Female BABL/c mice (6–8 weeks, 18–20 g) were purchased from the Experimental Animal Center of the Second Affiliated Hospital of Harbin Medical University. The animal study was reviewed and approved by the Animal Care and Use Committee of the Second Affiliated Hospital of Harbin Medical University (No. YJSDW 2022-126). Mice were raised in a specific pathogen-free condition (22 ±1  °C, relative humidity 55 ±1%, 12 h light/dark cycle) for at least 1 week before experiments. The mice were fed drinking water and standard laboratory chow ad libitum. As previously reported, imiquimod (IMQ) was used to construct a psoriasis-like mouse model (Huang et al., 2021; Gao et al., 2020). Mice were randomly divided into the IMQ group (n = 5) and the control group (n = 5). Mice were shaved on the back and treated with 5% IMQ cream (62.5mg, Mingxin, China) daily for 6 consecutive days. Control mice were treated with the same dose of vehicle cream (Vaseline, China). All mice were anesthetized by intraperitoneal injection of tribromoethanol (0.2ml/10g) at the beginning of the experiment. At the end of the experiment, mice were euthanized with pentobarbital sodium, and back skin was collected for RNA extraction.

Real-time quantitative polymerase chain reaction (RT-qPCR)

Skin tissue samples of humans and mice were collected by skin biopsy, immediately frozen in liquid nitrogen and stored at −80 °C. TRIzol reagent (Invitrogen, Waltham, MA, USA) was used to extract RNA from the skin tissues. Subsequently, cDNA was synthesized from 1 µg of total RNA with ReverTra Ace® qPCR RT Master Mix with gDNA Remover (Toyobo, Osaka, Japan) by S1000 Touch™ Thermal Cycler (Bio-Rad, Hercules, CA, USA). RT-qPCR was performed using SYBR Green Master (Roche, Basel, Switzerland) and C1000 Touch™ Thermal Cycler (Bio-Rad, Hercules, CA, USA) with the following program: 95 °C for 10 min, followed by 40 cycles at 95 °C for 15 s and 60 °C for 1min. All RT-qPCR was performed using the 2−ΔΔCT method to calculate of the relative gene expression. Primer sequences are listed in Table S2.

Statistical analysis

Statistical analyses were performed using R software (version 3.6.1; R Core Team, 2019) and GraphPad Prism 9 software. The experimental results are expressed as mean ± standard deviation (SD). Kruskal-Wallis test was performed to analyze variations between two groups in bioinformatics analysis, while Student’s t-tests were utilized in RT-qPCR data analysis. P<0.05 is considered as statistical significance.

Results

M6A-related DEGs

In total, 16 m6A-related DEGs were identified in the LS, including ALKBH5, RBM15, YTHDF3, IGF2BP3, HNRNPA2B1, YTHDC2, FTO, WTAP, LRPPRC, YTHDC1, IGF2BP2, YTHDF2, HNRNPC, ZC3H13, METTL14, and METTL3. The expression levels of these genes are shown in Fig. 2A. Correlation analysis was conducted between the m6A-related DEGs. We found that the positive correlation between the expression levels of LRPPRC and YTHDF3 was the highest one (Fig. 2B). As shown in Fig. 2C, a protein-protein interaction network of m6A-related DEGs was constructed and 85 protein-protein pairs were obtained. Further, GO and KEGG pathway enrichment analyses showed that these genes were significantly enriched in 77 biological processes (BPs), including regulation of mRNA metabolic processes, RNA catabolic processes, and regulation of RNA stability; six cellular components, including nuclear speck; 22 molecular functions, including catalytic activity acting on RNA; and five KEGG pathways, including the metabolism of RNA and processing of capped intron-containing pre-mRNA (Figs. 2D–2G). Meanwhile, 16 m6A-associated DEGs, which are involved in various RNA-associated biological processes, cellular components, molecular functions and KEGG pathways, were identified in LS.

Figure 2 M6A-related differentially expressed genes (DEGs) between psoriasis lesions (LS) and non-lesions (NL) and functional analyses.

(A) Distribution map of m6A-related DEGs. A total of 16 m6A-related DEGs were identified. The relative expression is shown as the mean ±SD. (B) The correlation between the expression levels of 16 m6A-related DEGs. The size of the numerical value represents the correlation coefficient, and the larger the numerical value, the higher the correlation. (C) Protein-protein interaction network of 15 m6A-related DEGs; the node color indicates significance; the size of the node indicates the significance of difference; the larger the difference, the higher the significance; the thickness of the connection indicates the size of the combined score, and the thicker the connection, the higher the difference. (D) Column distribution diagram of biological processes enriched in m6A-related DEGs. (E) Column distribution diagram of cellular components enriched in m6A-related DEGs. (F) Column distribution diagram of molecular functions enriched in m6A-related DEGs. (G) Column distribution diagram of KEGG pathways enriched in m6A-related DEGs.

Construction of a diagnostic model diagnosis based on m6A-related DEGs

As illustrated in Fig. 3A, all 16 m6A-related genes showed significantly different expression levels between the LS and NL groups. Based on the coefficients and mean-squared error, ten optimal m6A-related DEGs were identified, including FTO, IGF2BP2, METTL3, YTHDC1, ZC3H13, HNRNPC, IGF2BP3, LRPPRC, YTHDC2, and HNRNPA2B1 (Fig. 3B).

Figure 3 Selection of optimal m6A-related differentially expressed gene combinations.

(A) Single factor logistic regression forest map of 16 m6A-related differentially expressed genes. (B) LASSO filter parameter diagram.

The risk score of the samples in GSE30999 was calculated according to the LASSO coefficient. We found that the risk score of LS was significantly higher than that of NL (Fig. 4A). The ROC curve was constructed, and the area under the ROC curve (AUC) was 0.974 (0.918, 0.924), suggesting a high diagnostic value of the diagnostic model (Fig. 4B). The heatmap showed clearly distinguishable expression levels of these ten m6A-related DEGs in NL and LS (Fig. 4C).

Figure 4 Risk value distribution of diagnostic model, receiver operating characteristic curve (ROC), and heatmap of ten optimal m6A-related differentially expressed gene (DEG) combinations in GSE30999 and GSE13355.

(A) Risk value distribution of diagnostic model of ten optimal m6A-related DEGs in GSE30999. (B) ROC of ten optimal m6A-related DEGs combinations in GSE30999. (C) heatmap of ten optimal m6A-related DEGs combinations in GSE30999. (D) Risk value distribution of diagnostic model of ten optimal m6A-related DEGs in GSE13355. (E) ROC curve of ten optimal m6A-related DEGs combinations in GSE13355. (F) Heatmap of ten optimal m6A-related DEGs combinations in GSE13355.

For the verification dataset GSE13355, risk scores, ROC curves, and heatmap analyses based on the ten m6A-related DEGs were also performed. Similarly, a higher risk score for psoriasis was observed (Fig. 4D). The AUC of the diagnostic model was 0.730 (0.906, 0.707) (Fig. 4E), and the heatmap showed distinct expression levels of these ten m6A-related DEGs in NL and LS (Fig. 4F). Taken together, we identified ten optimal m6A-related DEGs that discriminated between LS and NL groups and constructed a diagnostic model with high accuracy and specificity based on their risk scores, ROC curves, and heatmaps in two independent datasets.

The network associated with m6A-related DEGs

Next, an investigation was conducted into the network that could potentially regulate m6A-related DEGs within the diagnostic model. As shown in Fig. 5A, 80 miRNA-m6A-related DEGs pairs were identified in the miRNA-m6A-related DEGs network. Among the ten m6A-related DEGs, YTHDC1, HNRNPC, and FTO were targeted by more miRNAs. Further, TF-m6A related genes network showed that there were 82 TF-m6A-related DEGs pairs and nine TFs (Fig. 5B).

Figure 5 Networks associated with ten optimal m6A-related differentially expressed genes (DEGs).

(A) miRNA-ten optimal m6A-related DEGs network; Circle and triangle represent m6A gene and miRNAs respectively. (B) Transcription factor (TF)-ten optimal m6A-related DEGs network; Circle and square represent m6A gene and TFs respectively. (C) Small molecular drugs- ten optimal m6A-related DEGs network; Circle and square represent drugs and m6A-related DEGs respectively. (D) Interaction network diagram based on GeneMANIA.

Meanwhile, 57 small molecule drugs-m6A-related DEGs pairs were identified in the small molecule drugs-m6A-related DEGs network, in which dorsomorphin and cyclosporine were targeted by more m6A-related genes (Fig. 5C). The network constructed by GeneMANIA showed that these genes were mainly enriched in functions such as RNA destabilization, regulation of RNA stability, and regulation of mRNA catabolic processes (Fig. 5D). Overall, those findings suggest that m6A-related DEGs were regulated by various miRNAs, TFs, and small molecule drugs.

Mast cell and NK cell activation widely related to all m6A-related DEGs

Compared to NL, the infiltration of 12 immune cells was significantly different in LS. Among the 12 immune cells, the levels of nine immune cells were significantly higher in LS, including activated myeloid dendritic cells, macrophage M1, and CD4 memory resting T-cells (Fig. 6A). The correlation between m6A-related DEGs and the 12 immune cells is shown in Fig. 6B. We found that the infiltration of the activated mast and NK cells were significantly related to all ten m6A-related DEGs. These results suggest that m6A-related DEGs may play important roles in modulating the immune response in LS.

Figure 6 The relationship between expression levels of m6A-related differentially expressed genes (DEGs) and the inflation of immune cells.

(A) Distribution of 22 immune cells in psoriasis lesions (LS) and non-lesional (NL) tissues. (B) heatmap of ten optimal m6A-related DEGs and 12 immune cells with differential distribution levels in LS and NL.

Analysis of psoriasis subtypes based onm6A-related DEGs

We aimed to further analyze the psoriasis sample based on ten diagnostic m6A-related DEGs. The LS samples from GSE30999 were divided into two subtypes: subgroups 1 and 2, including 63 and 22 samples, respectively (Fig. 7A). Furthermore, samples in subgroup 1 had significantly higher m6A scores than those in subgroup 2 (Fig. 7B). Except for three genes, including LRPPRC, METTL3, and HNRNPC, all other had similar expression levels genes in the two subgroups (Fig. 7C). Then, ROC curve analysis was used to evaluate the performance of the SVM model based on the expression levels of 10 genes categorized into two subtypes (Fig. 7D). The average area under the curve (AUC) of the molecular subtypes was 0.833, indicating the SVM model has good sensitivity and specificity. Next, we analyzed the differences in the infiltration of immune cells and expression levels of the HLA family genes between the two subtypes. The results revealed significant differences between subgroups 1 and 2 in the infiltration of two immune cells, including activated mast cells and follicular helper T cells. The ESTIMATE, immune, and strong scores were significantly higher in subgroup 2 than in subgroup 1 (Figs. 8A and 8B). Eight of 21 HLA family genes were significantly different between subgroups 1 and 2; these included TAP2, HLA DRA, HLA DMB, and TAP1 (Fig. 8C).

Figure 7 Subgroups based on ten optimal m6A-related DEGs.

(A) Cluster diagram of subtype analysis. (B) m6A scores between the two subtypes. The relative expression is shown as the mean ±SD. (C) Expression levels of ten optimal m6A-related DEGs in the two subgroups. The relative expression is shown as the mean ±SD. (D) ROC curve showing the performance of the SVM model.

Figure 8 Immune cell distribution and expression levels of HLA family genes in the two subgroups.

(A) Distribution of various immune cell types in the two subtype groups. The relative expression is shown as the mean ±SD. (B) Distribution of ESTIMATE, immune, and strong scores in different subtype groups. The relative expression is shown as the mean ±SD. (C) The expression levels of HLA family genes (10 genes) in the two subgroups. The relative expression is shown as the mean ±SD.

To clarify the biological differences between the two subgroups, we performed enrichment analyses of the differential gene expression. A total of 1,592 DEGs were obtained between subgroups 1 and 2. These DEGs were enriched in 115 GO BPs, including neutrophil activation involved in the immune response and neutrophil degranulation (Fig. 9A), and 23 KEGG pathways, including cytokine-cytokine receptor interaction and chemokine signaling pathway (Fig. 9B). Collectively, these results indicate that the two subgroups of LS samples have distinct molecular and immunological characteristics, which may have implications for the diagnosis and treatment of psoriasis.

Figure 9 GO biological process and KEGG pathways associated with differentially expressed genes between subgroup 1 and subgroup 2.

(A) GO biological process. (B) KEGG pathways.

Two modules significantly related to psoriasis subtypes

As shown in Fig. 10A, when the squared correlation coefficient reached 0.9 for the first time, power values were chosen (power =16). The average node connectivity of the co-expression network was 1, suggesting that the network was consistent with small-world networks. The dissimilarity coefficient between nodes was then calculated, and a hierarchical clustering tree was obtained. When the pruning height was 0.995 and the minimum number of genes in each module was 50, seven modules were obtained (Fig. 10B). The correlation between the phenotype (disease/control) of the sample and each module is shown in Fig. 10C. A total of 146 genes related to the yellow and green modules were defined as genes most closely related to the subtype groups. A protein-protein interaction network was further constructed, and 382 connection pairs were obtained with connection scores higher than 0.4 (Fig. 10D). Together, these results demonstrate that the co-expression network analysis can identify the key genes and modules associated with the disease phenotype and reveal the potential molecular mechanisms underlying the subtype groups.

Figure 10 Modules related to disease status based on the WGCNA algorithm.

(A) Adjacency matrix weight parameter power selection graph and schematic diagram of average connectivity of genes under different power parameters; The horizontal axis represents the weight parameter power, and the vertical axis represents the square of the correlation coefficient between log (k) and log (p (k)) in the corresponding network. The red line indicates the standard line when the squared correlation coefficient reaches 0.9. The red line in the right figure indicates the average connectivity of network nodes (1) under the power parameter of the adjacency matrix weight parameter in the left figure. (B) Tree diagram of the module, with each color representing different module. (C) Heatmap of correlation between each module and sample phenotype. (D) Protein-protein interaction network of the yellow and green modules. The triangle and inverted triangle represent upregulated genes in subgroups 1 and 2, respectively, and the edge color of nodes represents the color from the WGCNA module.

RT-qPCR validation of significant m6A genes

We verified three m6A genes in the diagnostic model with the threshold of p<0.05 and —log2FC—>0.5 in human and animal models. RT-qPCR results in the skin specimen showed that YTHDC2 was highly expressed in the psoriasis and IMQ group compared to the controls, while METTL3 and IGF2BP2 were decreased in the disease group (Fig. 11), which was consistent with the results in GSE30999.

Figure 11 RT-qPCR experimental validation of significant m6A genes.

(A–C) Relative mRNA expression of YTHDC2, METTL3 and IGF2BP2 in psoriasis and normal tissues ( n = 6). (D–F) Relative mRNA expression of YTHDC2, METTL3 and IGF2BP2 in the IMQ and control groups in mice ( n = 5). The experiments were performed in triplicate, and the relative mRNA expression is shown as the mean ±SD. *p < 0.05, **p < 0.01, ***p < 0.001, ****p < 0.0001.

Discussion

Psoriasis is a chronic, recurrent inflammatory skin disease with an increasing prevalence worldwide. It is associated with many important diseases, including psoriatic arthritis, depression, and metabolic syndrome (Branisteanu et al., 2022; Kovitwanichkanont, Chong & Foley, 2020). Therefore, early diagnosis of psoriasis is extremely urgent. Previous studies have demonstrated that big data combined with artificial intelligence can provide a more intuitive auxiliary diagnosis for patients with psoriasis (Tapak et al., 2021; Dash et al., 2020). In this study, m6A-related DEGs were explored to establish a model for the diagnosis of psoriasis. The diagnostic model based on ten m6A-related DEGs, including FTO, IGF2BP2, METTL3, YTHDC1, ZC3H13, HNRNPC, IGF2BP3, LRPPRC, YTHDC2, and HNRNPA2B1 showed relatively valuable role in the diagnosis of psoriasis.

It was found that there was a differential expression of m6A regulators in psoriasis (Liu et al., 2022). M6A regulators have been widely explored as potential diagnostic biomarkers for various diseases. For example, Liu et al. (2020) identified an m6A gene-based diagnostic signature for gastric cancer with high specificity and sensitivity. M6A transcriptome-wide profiling of psoriasis performed by Wang & Jin (2020) demonstrated a regulatory role of m6A in psoriasis. Our study demonstrated that the ten m6A-related gene signature might serve as a relatively valuable tool for psoriasis diagnosis, with an AUC of 0.974 (0.918, 0.924), suggesting that the signature-based ten m6A-related genes can effectively classify and diagnose psoriasis.

Cluster analysis divided the samples into two subgroups based on the expression of these m6A candidate molecules. The samples in subgroup 1 had significantly higher m6A scores than those in subgroup 2, and the expression levels of LRPPRC, METTL3, and HNRNPC differed significantly between subgroups. METTL3 is the core methyltransferase subunit, which can modulate the processing of RNA by recruiting specific “readers” proteins (Shi, Wei & He, 2019). Previous studies have shown that YTHDF1 and METTL3 upregulation is associated with poor survival in hepatocellular cancer (Shi et al., 2021). So far, among the m6A modulators, there have been the most studies on METTL3 and psoriasis. Several studies demonstrated that downregulation of METTL3 in psoriasis-lesional tissues and METTL3-mediated m6A modification promoted the proliferation of keratinocytes and inflammation (Xian et al., 2022; Wang, Huang & Jin, 2022). Xiao et al. found that METTL3 facilitates γ δ T17 cell-mediated proinflammatory response in imiquimod-induced psoriasis-like mouse model, and METTL3 is considered a potential target for treating γ δ T17 cell-mediated excessive inflammation (Xiao et al., 2023). The major roles of the pentatricopeptide repeat (PPR) family in RNA activation, including RNA editing, translation, and splicing, have been widely revealed. LRPPRC dysregulation is associated with the occurrence of various diseases, including viral infections and tumors. Several studies have shown that high LRPPRC levels are related to poor prognosis by affecting multiple pathways, including PI3K/AKT/mTOR signaling and autophagy (Cui et al., 2019). Moreover, previous evidence has shown that polymorphisms in the FTO gene can affect the severity of psoriasis (Merola et al., 2022). Meanwhile, compared to healthy controls, the mRNA level of IGF2BP3 in psoriasis lesions was upregulated (Xing et al., 2022), and the level of HNRNPA2B1 was significantly downregulated (Lu et al., 2021). Our experiments verified that the mRNA level of YTHDC2 was elevated in psoriasis patients, and the expression of METTL3 and IGF2BP2 was significantly decreased. The roles of other genes, including ZC3H13, HNRNPC and YTHDC1, in psoriasis have not yet been explored. However, they are also critical regulators of cellular activity and DNA damage (Mo et al., 2022; Wu et al., 2022). Altogether, in psoriasis, which is a multifactorial genetic disorder, it is recommended to further identify their roles in in vitro and in vivo studies.

Small-molecule drugs, including dorsomorphin and cyclosporine, were targeted by more m6A-related genes, suggesting that patients with psoriasis with m6A regulator modifications might benefit from these drugs. Dorsomorphin, also known as compound C, selectively inhibits bone morphogenetic protein (BMP) type I receptors. BMP inhibitors have been reported as promising new therapeutic targets for chronic inflammatory diseases, including psoriasis (Alvarez et al., 2020). Patients with psoriasis may benefit from these drugs.

As an immune-mediated disease, it is widely accepted that immune cells play an important role in the development and pathogenesis of psoriasis. Our data showed that mast cell and NK cell activation was significantly related to all ten m6A-related genes. As a type of hematopoietic cell, mast cells mainly reside in vascularized tissues and release proinflammatory cytokines upon appropriate activation, and this effect worsens with the development of psoriasis (Conti et al., 2019). In psoriatic skin lesions, inflammatory cytokines produced by NK cells are related to the pathogenesis of psoriasis, and recent genetic studies have confirmed their role in psoriasis (Dunphy & Gardiner, 2011). The activation of mast cells and NK cells may be a vital regulator of the immune response in psoriasis.

So far, this is the first study to report the diagnostic values of m6A regulators in psoriasis patients. However, this study has several limitations. First, a small number of patients were enrolled in the selected dataset, so a larger dataset is needed to verify our results further. Second, our findings should be verified using additional clinical data, several baseline characteristics such as age and family history, could also be related to the severity of psoriasis. Third, we focused on three significant genes among ten m6A-related DEGs in the diagnostic model, while the other genes also need to be confirmed by in vitro and in vivo experiments.

Conclusions

In summary, we successfully established a diagnostic model based on m6A-related DEGs, which could contribute to the early diagnosis and targeted treatment of psoriasis. The networks and immune cells associated with these m6A-related DEGs were also investigated. In our future studies, we will further analyze the diagnostic value of m6A-related DEGs and explore their function and mechanism in psoriasis.

Supplemental Information

Supplemental Information 1 MIQE checklist

Supplemental Information 2 Supplementarey Tables

Supplemental Information 3 Author checklist

Supplemental Information 4 Raw data

Supplemental Information 5 R script

We would like to thank the databases mentioned in our study. We would like to thank all participants involved in this study.

Additional Information and Declarations

Competing Interests

Author Contributions

Human Ethics

Animal Ethics

Data Availability

The authors declare there are no competing interests.

Jing Liu conceived and designed the experiments, performed the experiments, analyzed the data, prepared figures and/or tables, authored or reviewed drafts of the article, and approved the final draft.

Youlin Wang performed the experiments, analyzed the data, prepared figures and/or tables, authored or reviewed drafts of the article, and approved the final draft.

Yu Sheng performed the experiments, prepared figures and/or tables, and approved the final draft.

Limin Cai conceived and designed the experiments, analyzed the data, authored or reviewed drafts of the article, and approved the final draft.

Yongchen Wang conceived and designed the experiments, authored or reviewed drafts of the article, and approved the final draft.

The following information was supplied relating to ethical approvals (i.e., approving body and any reference numbers):

The ethics committee of the First Affiliated Hospital of Harbin Medical University approval to carry out he study within its facilities.

The following information was supplied relating to ethical approvals (i.e., approving body and any reference numbers):

The animal study was reviewed and approved by the Animal Care and Use Committee of the Second Affiliated Hospital of Harbin Medical University (YJSDW 2022-126).

The following information was supplied regarding data availability:

The raw measurements are available in the Supplementary Files.

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
