# Peer review of "Construction and validation of m6A-related diagnostic model for psoriasis"

_PeerJ, doi:10.7717/peerj.17027_

## Round 0.1 · original submission · Major Revisions

The manuscript needs revision. Please see detailed comments from the reviewers. Need update figures, rearrange the text, prepare clear conclusion and discussion sections.

Reviewer 1 ·

Basic reporting

1. I cannot see the small text clearly in Figures 1, 5, 6, 7, and 10. Please increase the size and resolution of this figure.
2. Please provide the version of the R packages or other software, which is beneficial to reproducibility.
3. In line 91, " M6A-related genes were first obtained from the literature". How many genes are related to M6A? Readers may be interested in the portion of M6A-related genes that are relevant to this disease.
4. What is the meaning of the value of Figure 2B? In Fig. 2B. there is an interaction value for every pair of genes. The author only mentioned " a heatmap of the expression levels of m6A-related DEGs in the LS is shown in Fig. 2B", which seems to be not matched to the Figure.
5. Gene expression level cannot be negative in Figures 4C and 4F. Please annotate some labels like "-log value" to the figures.
6. In line 241, the author mentions "The aim of investigating the networks that may be involved in regulating m6A-related DEGs in the diagnostic model. " This sentence is not complete. Please add what the aim is.
7. If the author can provide the R script to generate the result, it will be beneficial to reproducibility.

Experimental design

1. I suggest the author check the batch effect of each dataset before all analysis. If there is a batch effect, authors should preprocess the data. Differential genes analysis should be based on the dataset without batch effect.
2. The author did not perform cross-validation for the lambda selection in the LASSO model.
3. The authors use SVM for diagnosis. I suggest authors provide some interpretation (e.g., direct interpretation or using the SHAP tool, https://shap.readthedocs.io/en/latest/index.html) and check whether the interpretation result is consistent with differential gene expression.
4. The author found two subtypes of the psoriasis. The performance of SVM should be evaluated respectively for each subtype of psoriasis. This can ensure that the SVM can work well for both subtypes or it can only work well for one of the subtypes.

Validity of the findings

Please ensure that the discovered subtype is not due to batch effect.

Reviewer 2 ·

Basic reporting

Please see additional comments.

Experimental design

Please see additional comments.

Validity of the findings

Please see additional comments.

Additional comments

The authors conducted a series of bioinformatics analyses to explore m6A regulators that were predicted to be involved in psoriasis, thereby constructing a diagnostic model that could be useful for psoriasis diagnosis. The current study would be much improved if the authors address the following concerns:


------[Major Concerns about FIGURES, METHODS, RESULTS, and/or CONCLUSIONS]
1. In all FIGURES, it would be clear and more readable to BOTH provide figures with high resolution AND expand on figure legends by explaining the meanings of colors, groups, lines, and abbreviations. These revisions would greatly help readers to understand the results and their implications easily and efficiently. For example,
1.1 In all FIGURES' bar graphs, it would be more informative to display individual data points; in other words, please replace bar graphs by EITHER scatter plots with bars OR scatter plots (a pattern like PMID: 34537192, PMID: 37046252, and PMID: 37452367). Bar graphs have been shown to be misleading, because they cannot reveal variation/dispersion within data; instead, scatter plots with bars could be acceptable and scatter plots would be preferable (as confirmed by PMID: 25901488 and PMID: 28974579).
1.2 In all FIGURES' legends, it would be more rigorous to mention BOTH the sample size (the number of data points OR how many samples/patients were included) AND whether the data points were technical or biological replicates.
1.3 In all FIGURES' legends, it would be more rigorous to mention how the authors reported the data error (variation/dispersion): standard deviation (SD), confidence intervals (CI), or standard error of the mean (SEM, which would be not preferable).

2. In ABSTRACT, which could be rewritten:
2.1 In Methods ("GSE30999 and GSE13355 were downloaded as the training analysis dataset and validation dataset respectively"), it would be more informative and rigorous to mention the defining features of GSE30999 and GSE13355: what types of tissues and how many samples.
2.2 In Methods ("A series of analyses were carried out for the screening and construction of a diagnostic model ... including single-factor logistic regression analysis, LASSO algorithm, Support Vector Machine (SVM) method, weighted gene co-expression network analysis(WGCNA), Gene Ontology (GO) and Kyoto Encyclopedia of Genes and Genomes (KEGG) analysis, et al"), it would be clearer (easier to understand) to split this into multiple sentences — mention each method and its rationale (why the step/algorithm was used).
2.3 In Methods and/or Results, the authors did not seem to define what is "m6A-related DEGs". To fill this gap, please briefly mention how they picked out "m6A-related DEGs".
2.4 In Results ("The area under the ROC curve (AUC) of GSE30999 was 0.974 (0.918, 0.924) and that of GSE13355 was 0.730 (0.906, 0.707)"), it would be clearer to rewrite this sentence. AUC is a term used for a diagnostic model rather than a GSE dataset, so the idea of "area under the ROC curve (AUC) of GSE30999" seems ambiguous. Does it mean the AUC of the m6A-related DEGs that were picked out from GSE30999?
2.5 In Results ("Meanwhile, the expression level of m6A regulators verified by RT-qPCR was consistent with the bioinformatics results"), it would be more informative and clearer to rewrite this sentence AND expand on what were "the bioinformatics results". In the current manuscript, the authors did not explicitly mention any bioinformatics results before this sentence.
2.6 In Results ("The inûltration of activated mast cells and NK cells was significantly associated with all ten m6A-related DEGs in psoriasis. Among them, YTHDC1, HNRNPC, and FTO were targeted by most miRNAs and were regulated by nine related TFs. Therefore, patients may benefit from dorsomorphin and cyclosporine therapy"), the authors did not mention the immune filtration analysis, miRNA network construction, and drug prediction in Methods. Thus, it would be more informative to touch on these methods as well as why to use them in Methods.
2.7 In Results ("Between the two subgroups, 1592 DEGs were identified, including LRPPRC and METTL3"), the authors did not seem to mention what were the "two subgroups" and how they developed in either Results or Methods. Thus, it would be more informative and clearer to pinpoint these aspects in Methods and/or Results.
2.8 In Conclusions ("A diagnostic model based on ten m6A-related DEGs in patients with psoriasis was constructed, which may provide early diagnostic biomarkers and therapeutic targets for psoriasis"), the authors did not seem to mention what was the "diagnostic model" AND how the model was established. Thus, it would be more informative and clearer to expand on these two ideas in Methods and/or Results.

3. In RESULTS:
3.1 It would be clearer to end each paragraph in RESULTS with one sentence: "Together, these results suggest that ..." (a pattern like PMID: 37452367, PMID: 34715879, PMID: 34384362, PMID: 35965679, and PMID: 34537192), summarizing a paragraph AND highlighting the implications of all results in the paragraph.


------[Minor Concerns about writing]
1. Throughout the manuscript, it seems better to use Grammarly (https://www.grammarly.com/) to check & correct potential grammatical errors or typos. For example,
1.1 In Methods of ABSTRACT ("Finally, experimental validation of significant m6A regulators in the diagnostic model by real-time quantitative polymerase chain reaction (RT-qPCR)"), it seems better to change this sentence into "Finally, the expression of significant m6A regulators involved in the diagnostic model was examined by real-time quantitative polymerase chain reaction (RT-qPCR)."

2. In SUPPLEMENTAL MATERIALS, it seems better to add the English version of "Consent form". This revision would help international readers better understand the text.

3. In ABSTRACT:
3.1 In Background ("Therefore, we aimed to explore the potential role of m6A in the diagnosis of psoriasis and the associated molecular mechanisms"), it seems better to change this sentence into "Herein, we aimed to explore the potential role of m6A regulators in the diagnosis of psoriasis and predict molecular mechanisms by which m6A regulators impact psoriasis." After this revision, the sentence would be more informative and clearer (easier to understand).
3.2 In Results ("DEGs were enriched in biological processes related to neutrophil activation and KEGG pathways, including cytokine-cytokine receptor interactions and chemokine signaling pathways"), it seems better to change this sentence into "These DEGs were predicted to be involved in neutrophil activation, cytokine-cytokine receptor interactions, and chemokine signaling pathways." After this revision, the sentence would be more concise.

---

## Round 0.2 · accepted · Accept

Thanks for the updates and detailed answers to the reviewers. The manuscript has no more critical remarks.

Reviewer 1 ·

Basic reporting

All issues have been addressed.

Experimental design

All issues have been addressed.

Validity of the findings

All issues have been addressed.

Reviewer 2 ·

Basic reporting

Please see Additional Comments.

Experimental design

Please see Additional Comments.

Validity of the findings

Please see Additional Comments.

Additional comments

Thank the authors for responding to all of my comments. The current version has been much improved.